# AUTOMATED MOVIE GENERATION VIA MULTI-AGENT COT PLANNING

## ABSTRACT

Existing long-form video generation frameworks lack automated planning, requiring manual input for storylines, scenes, cinematography, and character interactions, resulting in high costs and inefficiencies. To address these challenges, we present `MovieAgent`, an automated movie generation via multi-agent Chain of Thought (CoT) planning. `MovieAgent` offers two key advantages: 1) We firstly explore and define the paradigm of automated movie/long-video generation. Given a script and character bank, our `MovieAgent` can generates multi-scene, multi-shot long-form videos with a coherent narrative, while ensuring character consistency, synchronized subtitles, and stable audio throughout the film. 2) `MovieAgent` introduces a hierarchical CoT-based reasoning process to automatically structure scenes, camera settings, and cinematography, significantly reducing human effort. By employing multiple LLM agents to simulate the roles of a director, screenwriter, storyboard artist, and location manager, `MovieAgent` streamlines the production pipeline. Experiments demonstrate that `MovieAgent` achieves new state-of-the-art results in script faithfulness, character consistency, and narrative coherence. Our hierarchical framework takes a step forward and provides new insights into fully automated movie generation.

## 1 INTRODUCTION

Automated movie generation creates long-form videos with consistent characters, synchronized subtitles, and audio, given a script synopsis and character bank. It involves automating narrative planning, scene structuring, and shot composition, replicating the hierarchical reasoning of real-world filmmaking. Most existing video generation research Blattmann et al. (2023a); Zhou et al. (2022); Ho et al. (2022a); Wu et al. (2023) still focuses on short video generation without structured narratives, such as diffusion-based models like Stable Video DiffusionBlattmann et al. (2023a), Video LDMBlattmann et al. (2023b), and I2VGen-XLZhang et al. (2023a). More recently, spatiotemporal transformer models, including SoraBrooks et al. (2024) and HunyuanVideo Kong et al. (2024), have demonstrated superior performance in generating high-quality short videos (within 10 seconds) with realistic visuals and smoother motion dynamics. Compared to short-video generation, the development of long-form video generation Wu et al. (2024b); Yin et al. (2023); Hu et al. (2024); Polyak et al. (2024) has been relatively slow and still faces many challenges, such as maintaining narrative coherence, character consistency, structured scene transitions, and synchronized audio. DreamFactoryXie et al. (2024) uses multi-agent systems and video generation models to synthesize keyframes, later expanded into long-form videos. Similarly, StoryAgentHu et al. (2024) employs multiple agents for customized storytelling video generation. However, these approaches are limited to basic long-video synthesis, lacking high-level planning and logically structured multi-scene narratives. They also fail to handle multi-object interactions, customization, and audio consistency, making them unsuitable for real-world applications. Thus, automated movie-level long-form generation remains an open challenge in the field.

Let us delve deeper into understanding what is essential and indispensable in the real-world movie production process, as shown in Figure 1 (a). In reality, real-world movie production is a **hierarchical** and **collaborative** process, involving multiple specialized roles: directors, screenwriters, storyboard artists, and cinematographers, who work together to maintain narrative coherence, character consistency, and structured scene transitions. Therefore, unlike short-video generation, movie level video generation is a complex process, including high-level cinematic themes and low-level cinemato-

Figure 1: **Comparison of Traditional and Automated Movie Production.** Traditional filmmaking requires manual planning, while `MovieAgent` automates script breakdown, scene planning, and shot design, enhancing efficiency and narrative coherence.

graphic parameters, making it difficult to solve with a single model like an LLM or video generation framework.

Inspired by the real-world movie production process, we introduce multi-agent systems to simulate the roles of different filmmaking professionals and implement a hierarchical reasoning framework. As shown in Figure 1 (b), we propose `MovieAgent`, a Multi-Agent CoT Planning framework that automatically structures and generates multi-scene, multi-shot videos with logical storylines, synchronized subtitles, and consistent character appearances. The key advantages of `MovieAgent` include: 1) **CoT-based Hierarchical Reasoning**. Unlike direct inference, which lacks structured and in-depth planning, CoT-based reasoning enables step-by-step, interpretable decision-making while recording the rationale behind each decision for use in subsequent steps. 2) **Near-zero Cost.** Compared to real-world movie production, which requires millions of dollars and several years to complete, AI-driven movie generation (`MovieAgent`) is virtually cost-free. 3) **Multi-Agent** for Automated Filmmaking. `MovieAgent` incorporates multiple specialized AI agents that simulate the roles of a director, screenwriter, and storyboard Artist. Therefore, the multi-agent framework can automatically decomposes a movie synopsis into structured acts, scenes, and shots, ensuring coherent plot development and seamless transitions, as shown in Figure 1 (b). These agents collaboratively enables precise control over both high-level cinematic themes and low-level cinematographic parameters simultaneously.

To summarize, the contributions of this paper are:

- We firstly explore and define the paradigm of automated movie/long-video generation. Given a script and character bank, our `MovieAgent` can generates multi-scene, multi-shot long-form videos with a coherent narrative, ensuring character consistency, synchronized subtitles.

- `MovieAgent` employs a hierarchical CoT-based multi-agent reasoning framework to automate scene structuring, camera settings, and cinematography, reducing human effort. With internal CoT reasoning, `MovieAgent` effectively decouples and designs cinematic elements, including narrative structure and shot/scene composition.

- Experiments demonstrate that `MovieAgent` achieves state-of-the-art performance in automated storytelling and movie generation. Specifically, it excels in character consistency and narrative coherence.

## 2 RELATED WORKS

### 2.1 VIDEO GENERATION

Recent advancements in video generation have significantly improved quality, with approaches spanning diffusion models Ho et al. (2022b); Chen et al. (2023b); Zhang et al. (2023a); Wu et al. (2024a); Zhao et al. (2024), and transformer frameworks Yan et al. (2021); Yang et al. (2024); Kong et al. (2024). Diffusion models excel at image and video synthesis by progressively refining noise into realistic outputs. VDM Ho et al. (2022b) pioneered the use of diffusion for video generation, introducing a spatio-temporall architecture to model frame dependencies. SVD Blattmann et al.

(2023a) further advanced this by leveraging pre-trained text-to-image models for video generation, significantly improving quality. Lavie Wang et al. (2024b) introduces a high-quality video generation framework using cascaded latent diffusion models. More recently, SORA Brooks et al. (2024) and Hunyuanvideo Kong et al. (2024) showcased a highly coherent video generation system using advanced latent diffusion. Transformers have also proven effective for modeling long-range dependencies. VideoPoet Kondratyuk et al. (2023) introduced a VQ-VAE-based approach, tokenizing video frames and modeling them with an autoregressive transformer. CogVideo Yang et al. (2024) enhanced text-to-video generation using hierarchical attention, improving efficiency. VideoPoet Kondratyuk et al. (2023) used a multimodal transformer to improve video-text understanding, enabling more controllable and expressive video synthesis. Despite advances, existing frameworks still rely on manual input for narrative planning and scene composition. Our `MovieAgent` addresses these limitations by a multi-agent, where agents simulate key filmmaking roles, enabling fully automated movie generation.

## 2.2 STORY VISUALIZATION

Story visualization, which generates coherent visual sequences from text, is crucial for automated movie generation. Early GAN-based methods, such as StoryGAN Li et al. (2019), focused on maintaining narrative consistency. With the rise of diffusion models, approaches like StoryDiffusion Zhou et al. (2024), Magic-Me Ma et al. (2024) and DreamVideo Wei et al. (2024) improved temporal coherence and motion dynamics of videos. Adapter-based techniques, including IP-Adapter Ye et al. (2023), ROICtrl Gu et al. (2024) and In-context LoRA Huang et al. (2024a), enabled efficient fine-tuning for personalized and character-consistent generation. Meanwhile, structured story-to-video frameworks like AutoStory Wang et al. (2024a) and Make-a-story Rahman et al. (2023) enhanced scene composition and transition planning. However, existing methods still lack automated high-level planning, often requiring manual intervention for cinematography, scene structuring. We introduces a multi-agent CoT-driven framework, enabling fully automated and coherent long-form movie generation.

## 2.3 LLM FOR VIDEO GENERATION

Recent advancements in LLM-driven video generation Zhu et al. (2023); Wu et al. (2024b) have improved narrative structuring and interactive storytelling. VideoDirectorGPT Lin et al. (2023) and VideoStudio Long et al. (2024) explored LLM-powered frameworks for scene composition, while Mora Yuan et al. (2024) enhanced video conceptualization for long-form coherence. For storyboarding and cinematic planning, DreamFactory Xie et al. (2024) and StoryAgent Hu et al. (2024) introduced LLM-based adaptive shot planning, reducing manual effort in camera control and character interactions. VideoGen-of-Thought Zheng et al. (2024a) leveraged CoT reasoning to improve multi-shot video consistency. Although these methods enhance narrative structuring and storytelling with LLMs, they still require manual intervention or lack character and audio customization. In this paper, we firstly propose automated movie/long-video generation with a hierarchical CoT reasoning framework, which, given a script, character photos, and audio samples, automates planning, scene structuring, and cinematography for a more coherent and customizable filmmaking process.

## 3 METHOD

### 3.1 TASK FORMULATION

Given a script synopsis $S$ and a character bank $C$, the goal of automated movie generation is to generate a long-form video $\widehat{\mathcal{V}}$ consisting of multiple scenes and shots while ensuring narrative coherence, character consistency, and audiovisual synchronization. Formally, the objective is to find an optimal mapping function:

$$\mathcal{F} : (S, C) \to \widehat{\mathcal{V}} \tag{1}$$

where character bank $C = \{[\text{char}_k, I_k, A_k]\}_{k=1}^{L}$, $L$ denotes the number of characters, and $\text{char}_k$ is the $k$-th character names in the character list. $I_k$ and $A_k$ denotes the portrait images and audio samples of the character. The function $\mathcal{F}(\cdot)$ refers to the automated movie generation function that

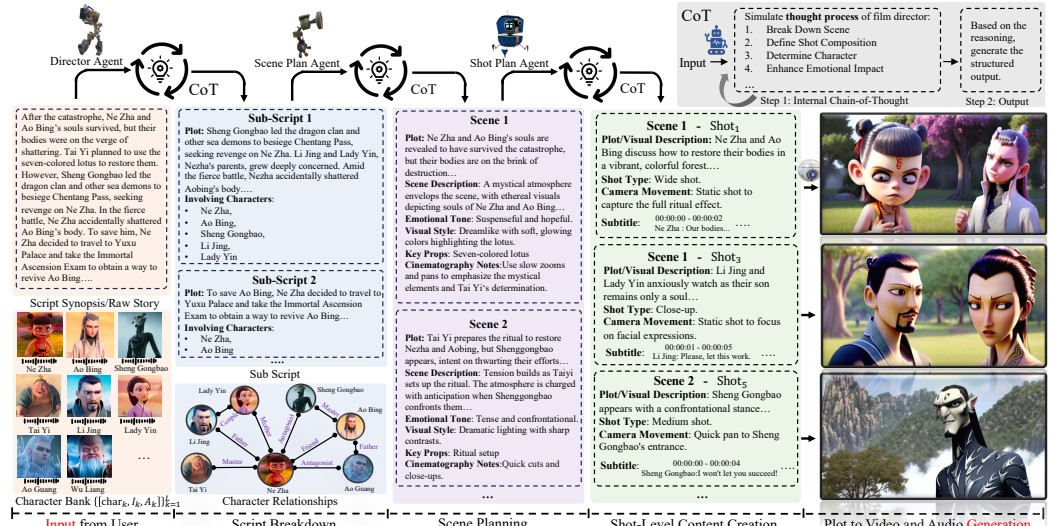

Figure 2: **The Overall Pipeline.** `MovieAgent` employs a hierarchical CoT reasoning process with director, scene plan, and shot plan agents to automate long-form movie generation.

systematically plans sub-scripts, scenes, and shots, along with various shot parameters, camera movements, and cinematographic settings. Ultimately, it generates a sequence of shots that collectively form the final movie output $\widehat{\mathcal{V}} = \{\widehat{V}_j^i \mid i = 1, 2, \ldots, N, \ j = 1, 2, \ldots, M\}$, where $\widehat{V}_j^i$ denotes the $j$-th shot video in the $i$-th scene.

## 3.2 AUTOMATED MOVIE GENERATION

`MovieAgent` leverages a multi-agent Chain of Thought reasoning process (§3.3) to achieve structured and automated movie generation, as shown in Figure 2. The system decomposes the filmmaking process into a **hierarchical workflow**, simulating key roles in traditional movie production. Specifically, we introduce three specialized agents: Director Agent (§3.2.1), Scene Plan Agent (§3.2.2), and Shot Plan Agent (§3.2.3), which collaboratively structure narratives, plan scenes, and generate detailed cinematographic elements. Then, customized shot and audio generation (§3.2.4) is utilized to produce the final audio and video.

### 3.2.1 DIRECTOR AGENT

The Director Agent is responsible for high-level narrative structuring. Given a script synopsis $S$ and a character bank $C$, it systematically decomposes the storyline into sub-scripts $\mathcal{S} = \{S_1, S_2, ..., S_K\}$, where each $S_p$ represents $p$-th key narrative unit that contributes to the overall plot development. The segmentation function can be formulated as:

$$\mathcal{S} = \mathcal{F}_{\text{Director}}(S, C, p), \quad p \in \{1, ..., K\} \tag{2}$$

where $\mathcal{F}_{\text{Director}}(\cdot)$ is the decomposition function that segments the script $S$ into meaningful sub-units $\mathcal{S}$ based on character interactions, thematic continuity, and narrative flow. Specifically, the director agent follows a structured reasoning process: *1) Identify Core Narrative Structure*: The director agent first analyzes the synopsis to identify main acts, key plot points, and turning points. *2) Define Script Segmentation*: Based on these core narrative elements, the script synopsis is divided into discrete, self-contained sub-scripts ($S_p$). *3) Ensure Logical Story Progression*: Each sub-script $S_p$ maintains temporal and thematic coherence across $\mathcal{S}$ for a cohesive plot. *4) Maintain Character Consistency*: The segmentation preserves the roles and relationships of characters from set $C$, ensuring their presence and interactions remain accurate throughout $\mathcal{S}$. *5) Justify the Division*: For each sub-script, a clear rationale for its segmentation (*e.g.,* major event shift, emotional climax, new setting introduction) must be provided, serving as a reference for the subsequent step-by-step reasoning process.

### 3.2.2 SCENE PLAN AGENT

With sub-scripts $\mathcal{S}$, the next step involves determining the movie scenes, key scene elements, and scene boundaries. The Scene Plan Agent is designed to refine sub-scripts $\mathcal{S}$ into scene compositions $\mathcal{P} = \{P_1, P_2, ..., P_N\}$, where each $P_i$ represents a detailed scene with enriched descriptions for the $i$-th scene. The scene planning process is formalized as:

$$\mathcal{P} = \sum_{p=1}^{K} \mathcal{F}_{\text{Scene}}(S_p, C, i), \quad i \in \{1, \dots, N\} \tag{3}$$

where $\mathcal{F}_{\text{Scene}}(\cdot)$ denotes the scene plan agent, which outputs a refined scene list $\mathcal{P}$. For the $i$-th scene $P_i$, the scene plan agent comprehensively summarizes factors such as involved characters, plot, emotional tone, visual style, and cinematography notes to thoroughly define the scene variables and expressive elements. Similar to the director agent, the scene plan agent follow a structured reasoning process: *1) Analyze the Narrative Structure*: The agent identifies key turning points and transitions, ensuring each scene forms a complete narrative with a clear start and end. *2) Extract Key Scene Elements*: The model identifies all characters, their roles, interactions, and key events in each major scene. *3) Define Scene Boundaries*: Finally, identify natural story breaks (*e.g.,* location shifts, time jumps, emotional climaxes), ensuring each scene has a clear purpose, and justify each division (*e.g.,* tone shift, new conflict). *4) Justify the Division*: Preserve the internal Chain-of-Thought behind scene segmentation and reasoning to ensure traceability and analyzability.

### 3.2.3 SHOT PLAN AGENT

Given the structured scenes $\mathcal{P}$, the shot plan agent is responsible for defining shot-level details, including character-aware plot, cinematographic parameters, and visual dynamics. Specifically, each scene $P_i$ is further decomposed into detailed shot compositions $\mathcal{V}^i = \{V_1^i, V_2^i, ..., V_M^i\}$, where each shot $V_j^i$ captures distinct visual perspectives and cinematographic intentions of the $j$-th shot video in the $i$-th scene. Therefore, with scene list $\mathcal{P} = \{P_1, P_2, ..., P_N\}$, the formalized function for shot-level decomposition is expressed as:

$$\mathcal{V} = \sum_{i=1}^{N} \mathcal{F}_{\text{Shot}}(P^i, C, j), \quad j \in \{1, \dots, M\} \tag{4}$$

where $\mathcal{F}_{\text{Shot}}(\cdot)$ is the Shot Plan Agent responsible for generating structured shots $\mathcal{V} = \{V_j^i \mid i = 1, 2, \dots, N, \ j = 1, 2, \dots, M\}$. Each shot level script $V_j^i$ includes rich, structured shot script, such as the involved characters, plot, camera movements, shot type, and character subtitles.

The Shot Plan Agent follows a structured reasoning workflow: *1) Determine Shot Composition and Framing*: Identify appropriate shot types (*e.g.,* wide, medium, close-up) and camera angles based on scene content and emotional impact. *2) Specify Cinematographic Techniques*: Clearly define camera movements (static, pan, tilt, zoom, tracking), lighting styles, and visual effects necessary. *3) Coordinate Visual Continuity*: Ensure visual coherence and consistency across shots within each scene, avoiding abrupt transitions or inconsistent visual styles. *4) Align with Scene Narrative*: Each shot should advance the narrative or enhance emotions, with reasoning for traceability and analysis.

### 3.2.4 CUSTOMIZED VIDEO AND AUDIO GENERATION

At this stage, given the shot level script annotation $V_j^i$ for the $j$-th shot in the $i$-th scene, the model invokes various customized image and video generation models (*e.g.,* AutoStory Wang et al. (2024a), StoryDiffusion Zhou et al. (2024), and Magic-Me Ma et al. (2024)) are used to produce the final shot-level video $\widehat{V}_j^i$.

Current video generation models, such as SVD Blattmann et al. (2023a), DreamVideo Wei et al. (2024) are unable to simultaneously support subtitle-to-audio generation. For the talking human generation task, some priors, such as Hallo2 Cui et al. (2024) and Edtalk Tan et al. (2024) primarily focuses on single human generation. Therefore, our current technology cannot fully address the simultaneous audio-video generation in a single model. Based on whether audio generation is required, we categorize the movie generation setting into two task:

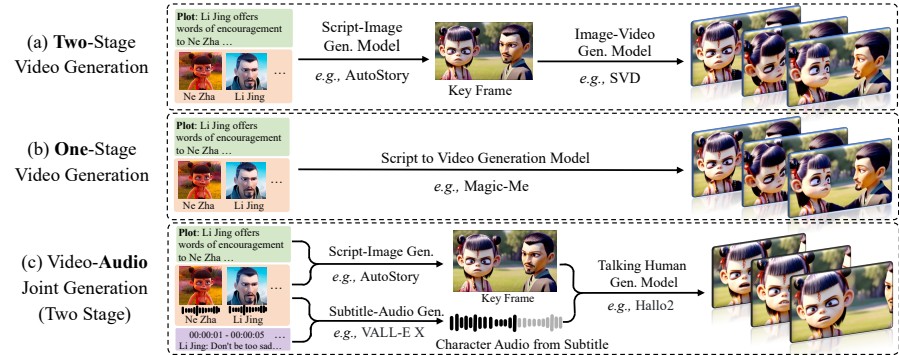

Figure 3: **Customized Shot-Level Video Generation for `MovieAgent`.** Current character-aware video generation can be divided into three categories: (a) Keyframe-based two-stage video generation; (b) One-stage end-to-end video generation; (c) Keyframe-based joint video and audio generation.

- **Pure Shot-level Video Generation in Figure 3 (a)-(b).** In this setting, we do not consider the audio generation of subtitle for the characters. Instead, we focus solely on generating pure video by modeling: $\widehat{\mathcal{V}}_j^i = \mathcal{F}_{\text{Video}}(V_j^i, C)$, where $\mathcal{F}_{\text{Video}}(\cdot)$ can either be a two-stage video generation model (*e.g.,* the combination of StoryDiffusion Zhou et al. (2024) and CogVideoX Yang et al. (2024)) or a one-stage customized end-to-end video generation model (*e.g.,* Magic-Me Ma et al. (2024)), as illustrated in Figure 3 (a)-(b).

- **Video and Audio Joint Generation in Figure 3 (c).** In this setting, the character bank $C$ include the voice sample of each character, represented as $\{[\texttt{char}_k, I_k, A_k]\}_{k=1}^L$. Since no model can simultaneously generate both audio and video, we adopt a two-stage video-audio generation strategy, as shown in Figure 3 (c). Formally, we express the formulation as: $\widehat{V}_j^i = \mathcal{F}_{\text{Talking}}\left(\mathcal{F}_{\text{Image}}(V_j^i, C), \mathcal{F}_{\text{Audio}}(V_j^i, C)\right)$, where $V_j^i$ includes subtitles for all characters. And $\mathcal{F}_{\text{Talking}}(\cdot)$, $\mathcal{F}_{\text{Image}}(\cdot)$, and $\mathcal{F}_{\text{Audio}}(\cdot)$ denote the talking-human generation model (*e.g.,* Hallo2 Cui et al. (2024)), customized image generation model (*e.g.,* StoryDiffusion Zhou et al. (2024)), and customized audio generation model (*e.g.,* VALL-E X Zhang et al. (2023b)), respectively.

## 3.3 INTERNAL CHAIN OF THOUGHT

The Internal Chain-of-Thought provides a general structured reasoning framework employed by various planning agents (*e.g.,* Scene Plan Agent, Shot Plan Agent, Director Agent) to methodically translate abstract narrative and cinematic requirements into detailed, actionable plans. Formally, given an input abstract $\mathbb{A}$ (*e.g.,* scene synopsis, script synopsis), the Internal CoT generates structured reasoning: $\mathbb{R} = \mathcal{F}_{\text{CoT}}(\mathbb{A})$, where $\mathcal{F}_{\text{CoT}}(\cdot)$ denotes the general internal reasoning function utilized by planning agents, involving explicit, systematic reasoning steps prior to the final generation of detailed cinematic output.

The Internal CoT generally encompasses the five stages: (1) Narrative Structure Analysis, which identifies major plot points, emotional beats, and key character interactions; (2) Key Element Extraction, focusing on essential characters, critical narrative events, and emotional significance; (3) Define Boundaries & Structural Units, establishing logical narrative divisions, scene transitions, and self-contained units; (4) Cinematic & Emotional Enhancement, refining visual style, emotional tone, lighting, props, and sound effects; and (5) Technical Cinematic Planning, specifying camera movements, shot compositions, character positions, and dialogues. Upon completion of this structured internal reasoning, agents produce a comprehensive, structured output that encapsulates all detailed planning elements necessary for subsequent execution by next stage.

## 4 EXPERIMENTS

### 4.1 EXPERIMENT SETTING

**Metric.** Following prior works Zheng et al. (2024a); Chen et al. (2023a), we evaluate the model using both automated metrics (*e.g.,* VBench Huang et al. (2023)) and human voting. Automated metrics

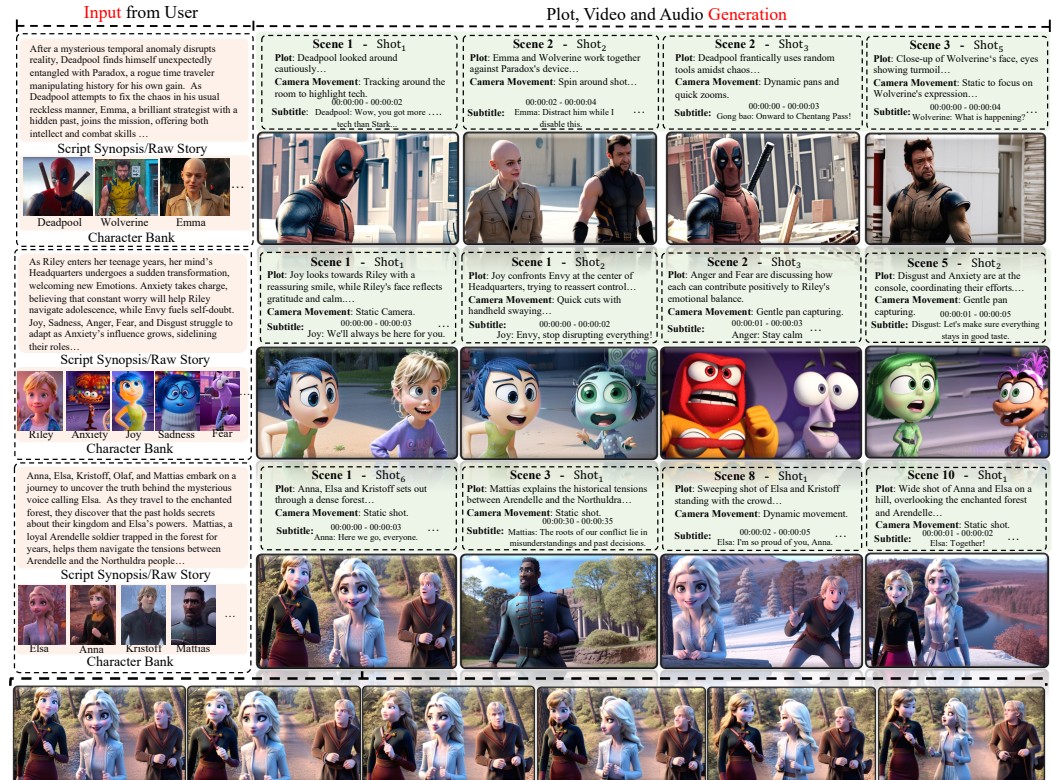

Figure 4: **More Visualizations for `MovieAgent`.** Our `MovieAgent` can generate coherent storylines and detailed shots.

Table 1: **Performance of Automatic metric for Script to Keyframe/Video Generation on `MovieAgent`.** Models without character consistency (*e.g.,*Open-Sora Zheng et al. (2024b)) are excluded. 'Subject Cons.', 'Bg Cons.', 'Motion Smth.', and 'Dyn. Degree' refer to 'Subject Consistency', 'Background Consistency', 'Motion Smoothness', and 'Dynamic Degree' from the advanced VBench Metrics Huang et al. (2023), respectively. GPT-4o serves as the LLM. And `MovieAgent` adopts multi-agent and internal CoT reasoning, while others rely on single-step generation.

| Method | CLIP↑ | Inception↑ | VBench Metircs Huang et al. (2024b)/% ↑ | | | | |
|---|---|---|---|---|---|---|---|
| | | | Subject Cons. | Bg Cons. | Motion Smth. | Dyn. Degree | Aesthetic |
| *Script Synopsis to Keyframe/StoryBoard Generation* | | | | | | | |
| StoryGen Liu et al. (2024) | 19.73 | 6.21 | - | - | - | - | - |
| StoryDiffusion Zhou et al. (2024) | 20.46 | 6.24 | - | - | - | - | - |
| AutoStory Wang et al. (2024a) | 20.21 | 6.01 | - | - | - | - | - |
| MovieAgent | **22.12** | **7.23** | - | - | - | - | - |
| *Script Synopsis to Movie Generation* | | | | | | | |
| StoryDiffusion + SVD | 21.39 | 8.36 | 93.64 | 93.78 | 96.30 | 74.48 | 56.69 |
| StoryDiffusion + CogVideoX | 21.83 | 9.01 | 93.45 | 94.56 | 96.60 | 27.89 | 56.05 |
| AutoStory + CogVideoX | 20.27 | 7.21 | 91.45 | 93.32 | 95.87 | 70.32 | 52.34 |
| DreamVideo | 21.37 | 8.11 | 93.17 | 93.77 | 96.40 | 26.97 | 42.16 |
| Magic-Me | 21.72 | 8.34 | 94.01 | **94.68** | 96.41 | 14.86 | 55.89 |
| MovieAgent | **22.25** | **9.39** | **94.72** | 93.52 | **97.84** | **76.27** | **58.63** |

offer objective analysis but may not fully match human preferences, while user studies capture real preferences but can be biased. Since automated movie generation varies in shot count and lacks a ground truth video, metrics like FID Heusel et al. (2017) cannot be computed. Two A6000 GPUs were used for all experiments.

**Baseline.** Existing approaches, such as DreamFactory Xie et al. (2024), StoryAgent Hu et al. (2024), and StoryDiffusion Zhou et al. (2024), fail to address the automated movie generation task (Section 3.1), struggling with multi-characters consistency and automated script planning. Therefore, we decompose the task into three components: LLM-based script processing (GPT4-o OpenAI (2025), Deepseek-R1 Guo et al. (2025), Llama3.3 Dubey et al. (2024)), image generation (AutoStory Wang

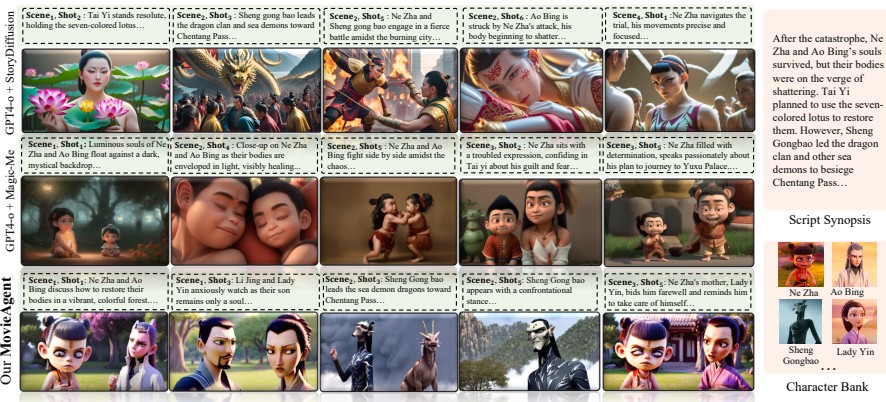

Figure 5: **Visualization Comparison for Different Methods.**

et al. (2024a),StoryDiffusion Zhou et al. (2024)), and video generation (DreamVideo Wei et al. (2024),Magic-Me Ma et al. (2024)). For each component, we incorporate baseline models for evaluation and comparison, as detailed in Table 1.

**Evaluation Dataset.** Since the automated movie generation task (Section 3.1) is formally defined for the first time, we need to construct a new evaluation dataset. This evaluation dataset takes as input a script summary, character names, photos, and audio samples, and outputs a series of shot videos. To achieve this, we propose a test set, namely MoviePrompts, consisting of 10 script prompts: 8 prompts are derived from well-known movies (*e.g.,* Ne Zha 2, Frozen II, and Inside Out 2), while the remaining 2 prompts (*e.g.,* Fictional stories and characters) are privately designed by two annotators.

### 4.2 PERFORMANCE COMPARISONS AND ANALYSIS

#### 4.2.1 AUTOMATIC METRIC ON MOVIEPROMPTS

Table 1 experimental results for script-to-keyframe and script-to-movie generation. In keyframe generation, `MovieAgent` achieves the highest CLIP score 22.12 and Inception scores 7.23, indicating superior visual-semantic alignment and image quality. For movie generation, `MovieAgent` consistently outperforms nearly all baselines across VBench metrics Huang et al. (2023), achieving the highest motion smoothness 97.84, dynamic degree 76.27, and aesthetic quality 58.63. These results highlight `MovieAgent` as a new state-of-the-art for automatic story-based video generation.

#### 4.2.2 HUMAN RATING

Figure 6 presents the user study for `MovieAgent` from two expert evaluators on the MoviePrompts dataset (10 movies). For a fairer and more fine-grained comparison, evaluators need rate each shot video on a scale of 1 to 5 based on the corresponding evaluation rules (detailed rules see supplementary materials). Due to the high cost of human ratings, we limited assessments to key baselines (GPT-4o with DreamVideo and Magic Me). `MovieAgent` present a promising performance, outperforming the best baseline by up to 2 points on a five-point scale. Notably, it excels in Narrative Coherence (3.49), Visual Appeal (4.01), Script Faithfulness (3.89), Character Consistency (4.04), and Physical Law (3.42). Figure 5 provides a relevant visual comparison, while Figure 4 presents additional visualizations, including coherent storylines, keyframes, and shot videos. These results highlight the effectiveness of our multi-agent and CoT reasoning approach.

### 4.3 ABLATION STUDY

#### 4.3.1 EFFECT OF INTERNAL CHAIN OF THOUGHT

Table 2 presents the ablation study for the internal Chain of Thought. Results show that GPT-4o with Internal CoT achieves a slight average score improvement (3.61 vs. 3.55 without CoT), with notable gains in Narrative Coherence (3.29 *vs.* 3.09). This is expected, as incorporating CoT enables

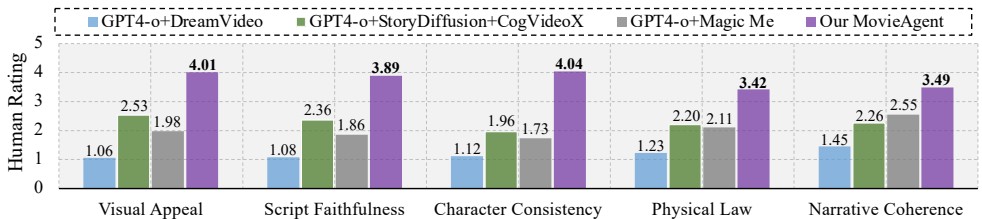

Figure 6: **Performance of User Study for Automated Movie Generation.** Under 0-5 rating system, `MovieAgent` demonstrates outstanding performance across script faithfulness and character consistency.

Table 2: **Ablation Study for LLM, CoT, and Multi-Agent.** In this experiment, ROICtrl Gu et al. (2024) and CogVideoX Yang et al. (2024) are used as the image and video generation methods. Due to the high cost of human evaluation (up to 100 shots per movie), our ablation study focuses on three movies: Ne Zha 2, Frozen II, and Inside Out 2.

| LLM Model | Internal CoT | Multi-Agent | Vis. Appeal | Script Faith. | Char. Consist. | Phys. Law | Narr. Coher. | Average |
|---|---|---|---|---|---|---|---|---|
| Llama3.3-70b | ✓ | ✓ | 3.86 | 3.50 | 4.00 | 2.70 | 3.13 | 3.44 |
| Deepseek-V3 | ✓ | ✓ | 3.74 | 3.66 | 3.78 | 3.07 | 3.45 | 3.54 |
| Deepseek-R1 | ✓ | ✓ | 4.02 | 3.59 | 4.09 | 3.38 | **3.79** | 3.78 |
| GPT4-o | | | 3.89 | 3.36 | 4.08 | 3.37 | 3.09 | 3.55 |
| GPT4-o | | ✓ | 4.02 | 3.69 | 4.13 | 3.46 | 3.31 | 3.72 |
| GPT4-o | ✓ | | 3.92 | 3.38 | 4.08 | 3.37 | 3.29 | 3.61 |
| GPT4-o | ✓ | ✓ | **4.04** | **3.92** | **4.11** | **3.49** | 3.55 | **3.82** |

step-by-step reasoning during story script generation, enhancing logical flow and coherence. By breaking down the reasoning process, CoT helps maintain narrative consistency and structure.

### 4.3.2 EFFECT OF LARGE LANGUAGE MODEL

Table 2 compares the performance of various LLMs: Llama3.3-70b, Deepseek-V3, Deepseek-R1, and GPT4-o across human evaluation including visual appeal, script faithfulness, character consistency, physical law, narrative coherence, and average. The results reveal that GPT4-o, particularly with multi-agent collaboration, achieves the highest average score of 3.82, outperforming Deepseek-V3 3.54, and Deepseek-R1 3.78. However, GPT-4o underperforms Deepseek-R1 in Narrative Coherence (3.55 vs. 3.79). This is expected, as Deepseek-R1 is optimized for reasoning tasks with a built-in Internal CoT process, enabling more extensive reasoning and generating smoother, richer narratives.

### 4.3.3 EFFECT OF MULTI-AGENT

Table 2 presents the evaluation for the impact of multi-agent collaboration. The results show that multi-agent collaboration significantly enhances performance, with GPT4-o achieving an average score of 3.72, compared to 3.55 without multi-agent collaboration. The improvements in script faithfulness and narrative coherence are particularly significant, with increases of 0.33 and 0.22 on a five-point scale, respectively. This is reasonable because multi-agent system is a hierarchical structure. Multi-agent collaboration is more efficient than single-step script generation, as it better translates a script synopsis into a full movie script with detailed plot logic.

## 5 CONCLUSION

In this paper, we firstly explore and define the paradigm of automated movie/long-video generation and propose `MovieAgent`, a multi-agent CoT-based framework for automated filmmaking. By integrating hierarchical reasoning and specialized AI agents, `MovieAgent` automates story structuring, scene planning, and shot composition, reducing human intervention while ensuring narrative coherence and cinematographic quality. Experiments show that `MovieAgent` improves story consistency, character preservation, and audiovisual synchronization, addressing key challenges in AI-driven filmmaking. Our approach provides a scalable solution for automated storytelling, offering new insights into the future of AI-assisted movie production.

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

## 6 CHECKLIST

### 6.1 THE USE OF LARGE LANGUAGE MODELS

In our work, LLMs are used for following aspects:

- Using an LLM to help with paper writing. We use GPT5 to help optimize language, correct grammar and write LaTeX table code.
- Using an LLM as a research assistant. We use GPT5 to help search related works.
- Using an LLM in our methods and experiment. This is described in the paper.

### 6.2 ETHICS STATEMENT

We confirm that our study did not use any sensitive data where all data are public available. We have conducted this research and reported our findings responsibly. All results are presented transparently, including both performance gains and any observed limitations. We have diligently cited all relevant prior work and data sources to give proper credit and context. By following best practices in documentation and research integrity, we aim to contribute positively to the scientific community while upholding the highest ethical standards.

### 6.3 REPRODUCIBILITY STATEMENT

We are committed to ensuring the reproducibility of our results. All code and data needed to reproduce the experiments will be made publicly available. We will release this repository openly with an appropriate open-source license upon publication. The datasets used in our experiments are standard public benchmarks for language modeling and understanding (e.g., widely-used corpora and evaluation sets). These resources are readily accessible to other researchers.

