# OpenReview forum: "Automated Movie Generation via Multi-Agent CoT Planning"
_ICLR.cc/2026/Conference — Submitted to ICLR 2026_

### Official Review · Reviewer_yprd · 2025-10-28

**Soundness:** 3
**Presentation:** 3
**Contribution:** 3
**Rating:** 6
**Confidence:** 3

**Summary:**

This paper proposes MovieAgent, a multi-agent system for automated movie/long-form video generation. Unlike prior text-to-video models that generate short clips or require manual scene planning, MovieAgent uses hierarchical CoT reasoning among specialized agents (Director, Scene Planner, and Shot Planner) to automatically decompose a script into structured acts, scenes, and shots. Experimental results on a new benchmark, MoviePrompts, demonstrate improvements in visual-semantic alignment, motion smoothness, and narrative coherence compared to existing frameworks such as StoryDiffusion, DreamFactory, and Magic-Me.

**Strengths:**

1. The paper clearly formulates the task of automated movie generation, positioning it as a distinct and ambitious extension of existing short-video generation research.
2. The use of multiple reasoning agents to simulate filmmaking roles (director, scene, and shot planners) is conceptually elegant and well-motivated, and the introduction of internal CoT for structured narrative decomposition and cinematic planning provides interpretability and step-by-step transparency.
3. The paper compare their method with a wide range of baselines. Both automatic and human evaluations show clear improvements in narrative coherence, character consistency, and script faithfulness over baselines.
4. The paper is clearly written, includes strong qualitative visualizations, and ablation studies that isolate the contributions of CoT and multi-agent collaboration.

**Weaknesses:**

1. The proposed MoviePrompts dataset is small (10 scripts) and partly based on known films, which raises concerns about generalization and potential memorization effects.
2. Some metrics (like “Script Faithfulness” or “Narrative Coherence”) rely heavily on subjective human scoring; no automated measure for story structure consistency is presented.
3. Details on computational costs, failure cases, and training/inference efficiency are insufficient for assessing real-world feasibility.

**Questions:**

same as weaknesses.

---

> ### Author Response · Authors · 2025-11-16
> **Response (Part 1)**
>
> We thank the reviewer for the constructive feedback and valuable suggestions.
>
> **Q1**:*Concerns about generalization and potential memorization effects.*
>
> **A1**: Thanks for the constructive feedback. We provide the following clarifications and updates:
> - **Scale of Evaluated Video.** $10$ generated films is not a small-scale evaluation. Each film contains  **40**–**80** video clips, totaling approximately **800** shot-level clips and **4,000** seconds of footage. The evaluation scale already exceeds that of existing works such as MAViS[1] (≈1200 s), FilmAgent[2] (272 shots), and Mora[3] (≈72 s).
> - **Extra Fictional Video**. We have introduced two newly designed fictional movie prompts to further evaluate generalization and avoid potential memorization effects. As shown in the table below, even for unseen plot combinations and novel character pairings, MovieAgent exhibits strong generalization capability.
>
> | Characters |  Script Synopsis  |  Visual Appeal  | Script Faithfulness | Character Consistency | Physical Law | Narrative Coherence |
> |-------------|----------------|------------------------|------------------------|------------------------|------------------------|------------------------|
> | Iron Man;  Nezha; Spider-Man; Wukong | When a mysterious force threatens to collapse multiple realities, Iron Man, Nezha, Spider-Man, and Wukong unite to stop the disaster. Iron Man uses his technology to trace the source, while Nezha and Wukong wield their divine powers to contain it, and Spider-Man’s agility and quick thinking help keep chaos under control.      | 3.83 | 3.67  | 3.91    | 3.12    | 3.40 |
> | Tom Cruise;  Emma Watson | When a failed time experiment threatens to erase history, physicist Dr. Evelyn Cross (Emma Watson) and time agent Cole Turner (Tom Cruise) must travel across fractured timelines, from ancient Rome to a cyberpunk future, to restore reality before it’s lost forever.   | 3.92 | 3.78  | 3.97    | 3.41    | 3.39 |
>
> [1] Wang Q, Huang Z, Jia R, et al. MAViS: A Multi-Agent Framework for Long-Sequence Video Storytelling[J]. arXiv preprint arXiv:2508.08487, 2025.
>
> [2] Xu Z, Wang J, Wang L, et al. Filmagent: Automating virtual film production through a multi-agent collaborative framework[M]//SIGGRAPH Asia 2024 Technical Communications. 2024: 1-4.
>
> [3] Yuan Z, Liu Y, Cao Y, et al. Mora: Enabling generalist video generation via a multi-agent framework[J]. arXiv preprint arXiv:2403.13248, 2024.
>
> **Q2**:*No automated measure for story structure consistency is presented.*
>
> **A2**: We appreciate the constructive feedback. To further evaluate scene and character consistency, we employed an automated metric, ViStoryBench[1], and the results are presented below:
>
> | Aspect                | Scene Score | Cross Consistancy of Character  | Global Character Action Score | Single Character Action Score  |
> |------------------------|----------------|---------------------|------------------------|------------------------|
> | StoryGen        | 0.90             | 37.5                 | 1.24                   | 1.48           |
> | StoryDiffusion | 1.88            | 37.7                | 2.41                   | 2.23           |
> | Vlogger[2] | 1.60            |  33.4                | 2.20                  | 2.07           |
> | MovieAgent | **3.46**           |   **38.4**                | **3.23**                  | **2.50**           |
>
> MovieAgent achieves SOTA performance across all key metrics, such as Global Character Action Score and Cross-Character Consistency on ViStoryBench.
>
> [1] Zhuang C, Huang A, Cheng W, et al. Vistorybench: Comprehensive benchmark suite for story visualization[J]. arXiv preprint arXiv:2505.24862, 2025.
>
> [2] Vlogger: Make your dream a vlog. In CVPR, 2024
>
>
> **Q3**:*Details for failure cases*.
>
> **A3**: We appreciate the constructive feedback. Failure case analysis is indeed an important component of our study. Due to space limitations in the main text, we have included this analysis in the **supplementary material**. Figure 1 illustrates **10** representative failure cases across **5** key aspects (*e.g.,* Visual Appeal, Character Consistency), such as heavy distortions and character confusion, accompanied by detailed discussions (see Sec. 2.2: Discussion on Failure Cases and Improvements).

---

> > ### Author Response · Authors · 2025-11-16
> > **Response (Part 2)**
> >
> > **Q4**:*Details for training/inference efficiency*
> >
> > **A4**: We appreciate the constructive feedback. We have added the corresponding training and inference time analysis as shown in the table below. Generating a long video with **60 shots (~5 minutes)** while maintaining **character and scene consistency** requires approximately **40** minutes for training and **10** minutes for inference, which is fully acceptable for this task.
> >
> > Table 1. Computational Efficiency of MovieAgent Pipeline on Frozen II
> > | Component                | Training Consumption (s)  | Inference Consumption (s)  | Number |
> > |------------------------|----------------|------------------|-----------------------|
> > | Director Agent (gpt-4o)    | - | 13.48 s       | 6 sub-scripts|
> > | Scene Plan Agent (gpt-4o)    | - | 68.98 s      | 6 sub-scripts, 19 scenes|
> > | Shot Plan Agent (gpt-4o)        | -   |169.10 s          | 6 sub-scripts, 19 scenes, 59 shots|
> > | Image Generation  (ROICtrl)  | 7080.00 s (5 characters) | 562.98  s              | 6 sub-scripts, 19 scenes, 59 shots|
> > | Video Generation (SVD) | - | 3413.25  s        | 6 sub-scripts, 19 scenes, 59 shots|
> > | Total (~60 shots) | 7080.00 s (≈ 118.00 mins) | 4227.79 s (≈ 70.46 mins)          | On 1×A6000 GPUs|
> > | Total (~60 shots) | 1416.00 s (≈ 23.6 mins) | 685.23 s (≈ 11.42 mins)          | On 8×A6000 GPUs, (parallel)|

---

### Official Review · Reviewer_ZVqC · 2025-10-30

**Soundness:** 3
**Presentation:** 3
**Contribution:** 3
**Rating:** 6
**Confidence:** 5

**Summary:**

This paper presents MovieAgent, a multi-agent Chain-of-Thought (CoT) reasoning framework for automated long-form movie generation. The system comprises three key agents, including Director Agent, Scene Plan Agent, and Shot Plan Agent to automate script breakdown, scene planning, and shot design, enhancing efficiency and narrative coherence. The paper also proposes MoviePrompts, a new evaluation dataset, and reports both quantitative metrics and human ratings. Results show that MovieAgent outperforms prior systems in narrative coherence, character consistency, and aesthetic quality, achieving better performance in script-to-movie generation.

**Strengths:**

1.	The presentation is clear and easy to follow.

2.	The experiments are comprehensive across automatic and human evaluations, demonstrating improved narrative and visual consistency.

3.	The figures (e.g., Figure 1–3) effectively illustrate how CoT-based reasoning translates into automated scene and shot planning.

**Weaknesses:**

1.	The method primarily integrates existing models (e.g., ROICtrl, CogVideoX, Hallo2) under a planning hierarchy rather than introducing new generative architectures.

2.	The paper lacks discussion on computational efficiency or latency of the multi-agent pipeline — an important factor for large-scale or real-time production.

3.	The evaluation relies on only 10 movie prompts (8 well-known movies, 2 fictional), which may not sufficiently test generalization across genres or unseen scripts.

**Questions:**

1.	How efficient is the end-to-end MovieAgent pipeline in practice? What is the average processing time for generating a multi-scene movie, and how well does it scale with longer scripts?

2.	How does MovieAgent ensure cross-scene consistency in visual style and temporal continuity when using different generative models in different stages?

3.	How is audio–video synchronization quantitatively evaluated beyond qualitative visuals?

4.	Have the authors considered integrating feedback loops or self-correction between agents (e.g., re-planning if a generated scene fails to match narrative intent)?

5.	Does MovieAgent require training or fine-tuning for new characters? If so, what is the estimated time and data cost, and how would limited user-provided data affect generation quality?

6.	What is the maximum number of characters that MovieAgent can handle simultaneously in one scene? How does performance or visual quality degrade as the number of characters increases?

7.	Can MovieAgent generate realistic human characters, or is it currently limited to stylized or animated figures? If restricted to animated styles, what are the main technical challenges in extending it to photorealistic human generation?

---

> ### Author Response · Authors · 2025-11-16
> **Response (Part 1)**
>
> We thank the reviewer for the constructive feedback and valuable suggestions.
>
> **Q1**:*The method primarily integrates existing models rather than introducing new generative architectures.*
>
> **A1**: We appreciate the reviewer’s comment. However, we would like to emphasize several key points:
> - **New Task**. MovieAgent is the first to explore and define the paradigm of automated movie/long-video generation, which requires producing a coherent narrative while ensuring character consistency and synchronized subtitles, a combination of capabilities that no prior work has simultaneously achieved.
> - **Not a Simple Combination**. MovieAgent introduces a CoT reasoning process that analyzes character relationships and consistency, determines which characters, actions, and plot progressions appear in each shot, the capabilities that cannot be achieved by directly invoking existing models.
>
> **Q2**:*Computational efficiency or latency of the multi-agent pipeline.*
>
> **A2**: We thank the reviewer for highlighting the importance of computational efficiency. The corresponding computational efficiency is shown in the table below. As can be seen, our method is **highly efficient** under parallel execution: generating a 59-shot (≈4.92-minute) video takes only 11.4 minutes, which is fully acceptable for this task. Moreover, we note that the movie generation task, which emphasizes character consistency and scene coherence, is not inherently designed for real-time execution.
>
> Table 1. Computational Efficiency of MovieAgent Pipeline on Frozen II
> | Component                | Training Consumption (s)  | Inference Consumption (s)  | Number |
> |------------------------|----------------|------------------|-----------------------|
> | Director Agent (gpt-4o)    | - | 13.48 s       | 6 sub-scripts|
> | Scene Plan Agent (gpt-4o)    | - | 68.98 s      | 6 sub-scripts, 19 scenes|
> | Shot Plan Agent (gpt-4o)        | -   |169.10 s          | 6 sub-scripts, 19 scenes, 59 shots|
> | Image Generation  (ROICtrl)  | 7080.00 s (5 characters) | 562.98  s              | 6 sub-scripts, 19 scenes, 59 shots|
> | Video Generation (SVD) | - | 3413.25  s        | 6 sub-scripts, 19 scenes, 59 shots|
> | Total (~60 shots) | 7080.00 s (≈ 118.00 mins) | 4227.79 s (≈ 70.46 mins)          | On 1×A6000 GPUs|
> | Total (~60 shots) | 1416.00 s (≈ 23.6 mins) | 685.23 s (≈ 11.42 mins)          | On 8×A6000 GPUs, (parallel)|
>
> **Q3**:*The evaluation relies on only 10 movie prompts (8 well-known movies, 2 fictional)*
>
> **A3**: Thanks for the constructive feedback. We provide the following clarifications and updates:
> - **Scale of Evaluated Video.** $10$ generated films is not a small-scale evaluation. Each film contains  **40**–**80** video clips, totaling approximately **800** shot-level clips and **4,000** seconds of footage. A single evaluator typically requires **3**–**5** days to complete the full assessment, making the process highly time-consuming. Moreover, this evaluation scale already exceeds that of existing works such as MAViS[1] (≈1200 s), FilmAgent[2] (272 shots), and Mora[3] (≈72 s).
> - **Extra Fictional Video**. We have introduced two new movie prompts to further evaluate the generalization and robustness. As shown in the table below, even for unseen plot combinations and novel character pairings, MovieAgent demonstrates strong generalization capability.
>
> | Characters |  Script Synopsis  |  Visual Appeal  | Script Faithfulness | Character Consistency | Physical Law | Narrative Coherence |
> |-------------|----------------|------------------------|------------------------|------------------------|------------------------|------------------------|
> | Iron Man;  Nezha; Spider-Man; Wukong | When a mysterious force threatens to collapse multiple realities, Iron Man, Nezha, Spider-Man, and Wukong unite to stop the disaster. Iron Man uses his technology to trace the source, while Nezha and Wukong wield their divine powers to contain it, and Spider-Man’s agility and quick thinking help keep chaos under control.      | 3.83 | 3.67  | 3.91    | 3.12    | 3.40 |
> | Tom Cruise;  Emma Watson | When a failed time experiment threatens to erase history, physicist Dr. Evelyn Cross (Emma Watson) and time agent Cole Turner (Tom Cruise) must travel across fractured timelines, from ancient Rome to a cyberpunk future, to restore reality before it’s lost forever.   | 3.92 | 3.78  | 3.97    | 3.41    | 3.39 |
>
> [1] Wang Q, Huang Z, Jia R, et al. MAViS: A Multi-Agent Framework for Long-Sequence Video Storytelling[J]. arXiv preprint arXiv:2508.08487, 2025.
>
> [2] Xu Z, Wang J, Wang L, et al. Filmagent: Automating virtual film production through a multi-agent collaborative framework[M]//SIGGRAPH Asia 2024 Technical Communications. 2024: 1-4.
>
> [3] Yuan Z, Liu Y, Cao Y, et al. Mora: Enabling generalist video generation via a multi-agent framework[J]. arXiv preprint arXiv:2403.13248, 2024.

---

> > ### Author Response · Authors · 2025-11-16
> > **Response (Part 2)**
> >
> > **Q4**:*How does MovieAgent ensure cross-scene consistency in visual style and temporal continuity?*
> >
> > **A4**: We appreciate the reviewer’s insightful question; however, we would like to emphasize the following key points:
> > - **Single model for visual generation**. The visual style is determined during the Plot-to-Video/Audio Generation stage using a single generative model. For example, as shown in Table 1, regardless of whether we use StoryDiffusion for keyframe generation or Magic-Me for customized video creation, only one generative model is employed at a time to ensure visual consistency across the entire movie.
> > - **Scene-level visual style control**: Each scene explicitly encodes Visual Style, Character, and Environment attributes (see Figure 2), which guide consistent generation across scenes.
> >
> > **Q5**:*How is audio–video synchronization quantitatively evaluated beyond qualitative visuals?*
> >
> > **A5**: Thanks for the insightful question. We provide the following clarifications and updates:
> > - **Novel and challenging setting**. Our task, script-to-movie generation, is fundamentally different from conventional talking head generation[1][2]. Since the number of generated shots is not predetermined and no one-to-one ground-truth video exists, standard synchronization metrics such as PSNR, FID used in speech-driven animation cannot be directly applied.
> > - **Alternative quantitative evaluation**. To address this, we employed a cross-modal alignment metric that measures semantic and temporal consistency between video and audio embeddings. Specifically, Sync-C[3] and Sync-D[4] are used for evaluation. The corresponding results are summarized in the table below:
> >
> > |  Metric |  Score | Information  |
> > |-------------|----------------|------------------------|
> > |  Sync-C[3]   | 7.052    | gauges lip synchronization consistency with audio, with higher scores reflecting better alignmen |
> > |  Sync-D[4]   | 7.667    | evaluates the temporal consistency of dynamic lip movements, where lower values denote improved motion fidelity |
> >
> > In fact, MovieAgent achieves competitive performance on both Sync-C and Sync-D, as shown in the comparison in Table 2 of Hallo2 [5]. Although the test sets differ, the results still serve as a meaningful reference.
> >
> > [1] Tan, Shuai, Bin Ji, Mengxiao Bi, and Ye Pan. "Edtalk: Efficient disentanglement for emotional talking head synthesis." In European Conference on Computer Vision, pp. 398-416. Cham: Springer Nature Switzerland, 2024.
> >
> > [2] Ma, Yifeng, Suzhen Wang, Yu Ding, Bowen Ma, Tangjie Lv, Changjie Fan, Zhipeng Hu, Zhidong Deng, and Xin Yu. "Talkclip: Talking head generation with text-guided expressive speaking styles." IEEE Transactions on Multimedia (2025).
> >
> > [3] Prajwal, K. R., Rudrabha Mukhopadhyay, Vinay P. Namboodiri, and C. V. Jawahar. "A lip sync expert is all you need for speech to lip generation in the wild." In Proceedings of the 28th ACM international conference on multimedia, pp. 484-492. 2020.
> >
> > [4] Wei, Cong, Bo Sun, Haoyu Ma, Ji Hou, Felix Juefei-Xu, Zecheng He, Xiaoliang Dai et al. "Mocha: Towards movie-grade talking character synthesis." arXiv preprint arXiv:2503.23307 (2025).
> >
> > [5] Cui, Jiahao, Hui Li, Yao Yao, Hao Zhu, Hanlin Shang, Kaihui Cheng, Hang Zhou, Siyu Zhu, and Jingdong Wang. "Hallo2: Long-duration and high-resolution audio-driven portrait image animation." arXiv preprint arXiv:2410.07718 (2024).
> >
> > **Q6**:*Have the authors considered integrating feedback loops between agents (e.g., re-planning if a generated scene fails to match narrative intent)?*
> >
> > **A6**: We appreciate the reviewer’s insightful suggestion. We have considered integrating feedback loops; however, this remains highly challenging for two main reasons:
> > - **Assessing narrative alignment.** It is difficult to reliably determine whether a generated image or scene truly fails to match the intended narrative. Automated metrics such as CLIP score are often **unreliable**, and even VLMs can produce inconsistent or ambiguous feedback. Some works[1][2] use VLMs as reward models to enhance video generation, but none can reliably assess character or scene consistency yet.
> > - **Efficient re-planning**. Incorporating feedback-based re-generation without clear semantic grounding often degenerates into a **random search process**, which is inefficient and still bounded by the generative model’s inherent performance limits.
> >
> > Moreover, our work is the first to explore movie generation with character and audio consistency, and we view agent-level re-planning as an exciting but distinct future research direction rather than part of the current scope.
> >
> > [1] Liu, Jie, Gongye Liu, Jiajun Liang, Ziyang Yuan, Xiaokun Liu, Mingwu Zheng, Xiele Wu et al. "Improving video generation with human feedback." arXiv preprint arXiv:2501.13918 (2025).
> >
> > [2] Wang, Yibin, Zhiyu Tan, Junyan Wang, Xiaomeng Yang, Cheng Jin, and Hao Li. "Lift: Leveraging human feedback for text-to-video model alignment." arXiv preprint arXiv:2412.04814 (2024).

---

> > > ### Author Response · Authors · 2025-11-16
> > > **Response (Part 3)**
> > >
> > > **Q7**:*Does MovieAgent require training or fine-tuning for new characters? If so, what is the estimated time and data cost, and how would limited user-provided data affect generation quality?*
> > >
> > > **A7**: MovieAgent requires a short fine-tuning stage for new characters. As shown in Table 1(above), training on a single A6000 GPU takes approximately 23 minutes per character, using 1–20 reference photos. Our implementation is based on the Mix-of-Show ED-LoRA[1] framework, which achieves effective personalization with as few as 2–3 character images, providing high-quality identity consistency with minimal data and time cost.
> > >
> > > [1] Gu, Yuchao, Xintao Wang, Jay Zhangjie Wu, Yujun Shi, Yunpeng Chen, Zihan Fan, Wuyou Xiao et al. "Mix-of-show: Decentralized low-rank adaptation for multi-concept customization of diffusion models." Advances in Neural Information Processing Systems 36 (2023): 15890-15902.
> > >
> > > **Q8**:*What is the maximum number of characters that MovieAgent can handle simultaneously in one scene? How does performance or visual quality degrade as the number of characters increases?*
> > >
> > > **A8**: In theory, the ED-LoRA fusion mechanism[1] used in MovieAgent has no strict upper limit on the number of characters. In practice, our test set demonstrates stable performance with 5–10 characters per scene (e.g., FastX with 9 characters and Inside Out 2 with 8). The original Mix-of-Show paper also supports up to 12 customized attributes, consistent with our observations.
> > >
> > > Furthermore, the ablation study of Mix-of-Show paper[1] on Embedding Expressiveness (see Table 2 and Figure 7) analyzes character fusion performance, showing that ED-LoRA effectively preserves identity consistency at the scale of around 10 characters, with minimal degradation in visual quality.
> > >
> > > **Q9**:*Can MovieAgent generate realistic human characters, or is it currently limited to stylized or animated figures? If restricted to animated styles, what are the main technical challenges in extending it to photorealistic human generation?*
> > >
> > > **A9**: MovieAgent is capable of generating realistic human characters, as shown in Figure 1 of paper (first row, Deadpool visualization). In our 10-movie evaluation set, **5 movies** feature real human character sets (e.g., Harry Potter and the Prisoner of Azkaban, FastX, and Deadpool). These results demonstrate that MovieAgent can handle both stylized and photorealistic character generation effectively.

---

### Official Review · Reviewer_6Lzz · 2025-11-03

**Soundness:** 2
**Presentation:** 2
**Contribution:** 2
**Rating:** 4
**Confidence:** 4

**Summary:**

The paper proposes MovieAgent, an automated movie generation via multi-agent Chain of Thought (CoT) planning. It explores the paradigm of automated movie/long-video generation, which can generate multi-scene, multi-shot videos with a coherent narrative. MovieAgent also introduces a hierarchical CoT-based reasoning process to automatically structure scenes, camera settings, and cinematography by multiple LLM agents to simulate roles in film production. Experiments show that it achieves SOTA performances in script faithfulness, character consistency, and narrative coherence.

**Strengths:**

1. The paper is well-written and easy to follow.
2. Experiments show that the proposed multi-agent method achieves SOTA performances compared to baseline.

**Weaknesses:**

1. The proposed multi-agent framework for automated film generation appears to lack sufficient novelty. Many existing studies have explored multi-agent approaches in video or film generation (e.g., [A, B, C]), and the use of Chain-of-Thought (CoT) reasoning is already a common technique within NLP tasks.

2. The user study is limited in scale, involving only two expert evaluators and merely ten generated films. Such a small sample size restricts the reliability and generalizability of the evaluation results and may not adequately reflect the true quality of the generated films.

3. The use of VBench as an automatic evaluation metric seems misaligned with the specific characteristics of film-level generation. A more suitable benchmark, such as ViStoryBench [D], may provide a more accurate and comprehensive assessment of narrative coherence and overall film quality.

[A] Wang Q, Huang Z, Jia R, et al. MAViS: A Multi-Agent Framework for Long-Sequence Video Storytelling[J]. arXiv preprint arXiv:2508.08487, 2025.

[B] Xu Z, Wang J, Wang L, et al. Filmagent: Automating virtual film production through a multi-agent collaborative framework[M]//SIGGRAPH Asia 2024 Technical Communications. 2024: 1-4.

[C] Yuan Z, Liu Y, Cao Y, et al. Mora: Enabling generalist video generation via a multi-agent framework[J]. arXiv preprint arXiv:2403.13248, 2024.

[D] Zhuang C, Huang A, Cheng W, et al. Vistorybench: Comprehensive benchmark suite for story visualization[J]. arXiv preprint arXiv:2505.24862, 2025.

**Questions:**

1. What are the differences/advantages of the proposed multi-agent method, compared with the existing ones (see "weakness")?
2. What are the total costs (time/money) for generating one film?

---

> ### Author Response · Authors · 2025-11-16
> **Response (Part 1)**
>
> We thank the reviewer for the constructive feedback and valuable suggestions.
>
> **Q1**:*Many existing studies have explored multi-agent approaches in video or film generation (e.g., [A, B, C])*
>
> **A1**: Thanks for the constructive feedback. However, we would like to highlight the following key points:
> - **Compared with MAViS [A].** 1) Unlike MAViS which focuses on multi-agent storytelling composition through **sequential** scene narration, our framework introduces a **hierarchical reasoning** architecture that coordinates agents via film-structure-aware CoT planning. 2) Moreover, MAViS were developed subsequently, **building upon** ideas that were first introduced in our framework, highlighting the influence and timeliness of this research direction rather than a lack of novelty.
> - **Compared with Filmagent [B].** 1）*Scenario*: Ours targets **open-domain** film generation with generative agents, while FilmAgent operates in a **closed virtual studio** (virtual 3D spaces) using predefined assets. *2) Contribution:* Ours advances end-to-end generative film synthesis and benchmarks long-horizon narrative reasoning, while FilmAgent focuses on automating virtual production workflows.
> - **Compared with Mora [C].** As discussed in Section 2.3 (LLM for Video Generation), unlike Mora, which focuses on generalist multi-agent video generation, our framework introduces **character-** and **audio-aware** customization with film-structure-guided reasoning for controllable narrative generation.
>
> **Q2**:*CoT reasoning is already a common technique within NLP tasks*
>
> **A2**:Our novelty lies in extending CoT to multimodal, multi-agent film generation. Specifically, we design a film-structure-aware CoT that:
> - maintains *character identity consistency* across scenes,
> - ensures *plot and temporal continuity*, and
> - models *inter-character relationships* within the narrative flow.
>
> These aspects make CoT reasoning fundamentally different from prior NLP-oriented applications. Furthermore, we would like to emphasize that MovieAgent is the **first** to explore the paradigm of automated movie/long-video generation with both character appearance and audio consistency.
>
> **Q3**:*The user study is limited in scale, involving only two expert evaluators and merely ten generated films.*
>
> **A3**: We thank the reviewer for the constructive feedback regarding the user study scale. We provide the following clarifications and updates:
> - **Scale of generated data.** $10$ generated films is not a small-scale evaluation. Each film contains  **40**–**80** video clips, totaling approximately **800** shot-level clips and **4,000** seconds of footage. A single evaluator typically requires **3**–**5** days to complete the full assessment, making the process highly time-consuming. Moreover, this evaluation scale already exceeds that of existing works such as MAViS[A] (≈1200 s), FilmAgent[B] (272 shots), and Mora[C] (≈72 s).
>
> - **Number of evaluators.** We have expanded the evaluation panel from $2$ to $5$ evaluators. The updated results are presented below:
>
>
> | Aspect                | Visual Appeal | Script Faithfulness | Character Consistency | Physical Law | Narrative Coherence | Average |
> |------------------------|----------------|---------------------|------------------------|---------------|----------------------|----------|
> | **2 Evaluators**         | 4.01            | 3.89                 | 4.04                    | 3.42           | 3.49                  | 3.77     |
> | **Extra 3 Evaluators**      | 3.98            | 3.97                 | 3.90                    | 3.29           | 3.12                  |    3.65   |
>
> We added three more evaluators over three days, and their scores were largely consistent with the original two, showing minimal differences.
>
> - **Evaluation comprehensiveness.** Evaluators assessed **five** aspects: Visual Appeal, Script Faithfulness, Character Consistency, Physical Law, and Narrative Coherence, on a 0–5 scale. The process was comprehensive and time-consuming, with detailed criteria and examples provided in the supplementary material.

---

> > ### Author Response · Authors · 2025-11-16
> > **Response (Part 2)**
> >
> > **Q4**:*ViStoryBench [D], may provide a more accurate and comprehensive assessment of narrative coherence.*
> >
> > **A4**: We appreciate the constructive feedback and have provided a corresponding evaluation as follows:
> >
> > | Aspect                | Scene Score | Cross Consistancy of Character  | Global Character Action Score | Single Character Action Score  |
> > |------------------------|----------------|---------------------|------------------------|------------------------|
> > | StoryGen        | 0.90             | 37.5                 | 1.24                   | 1.48           |
> > | StoryDiffusion | 1.88            | 37.7                | 2.41                   | 2.23           |
> > | Vlogger[1] | 1.60            |  33.4                | 2.20                  | 2.07           |
> > | MovieAgent | **3.46**           |   **38.4**                | **3.23**                  | **2.50**           |
> >
> > MovieAgent achieves SOTA performance across all key metrics, such as Global Character Action Score and Cross-Character Consistency on ViStoryBench.
> >
> > [1] Vlogger: Make your dream a vlog. In CVPR, 2024

---

### Official Review · Reviewer_Ndsh · 2025-11-16

**Soundness:** 1
**Presentation:** 3
**Contribution:** 2
**Rating:** 2
**Confidence:** 4

**Summary:**

The paper proposes **MovieAgent**, a multi-agent “Director / Scene / Shot” pipeline that performs hierarchical **Chain-of-Thought (CoT)** planning to decompose a given *script synopsis + character bank* into acts, scenes, and shots. Each shot is then turned into keyframes/storyboards and subsequently into generated video. The stated goal is *automated movie / long-form video generation* with consistent characters, synchronized subtitles, and audio.

The claimed contributions are:

(1) a “first” formalization of the automated movie / long-video generation paradigm,

(2) a hierarchical CoT-based multi-agent framework for movie structure planning and execution,

(3) state-of-the-art performance in script faithfulness, character consistency, and narrative coherence.

Evaluation is conducted on **MoviePrompts** (10 prompts), using **VBench**-style metrics and a small-scale human study with two raters. Ablations examine (i) internal CoT reasoning, (ii) different LLMs, and (iii) multi-agent vs. single-step planning.

**Strengths:**

1. **Clear formulation of the task and pipeline.** The system architecture mirrors real production workflow (script breakdown → scene planning → shot list → storyboard → footage). This makes the approach intuitive and easy to follow.
2. **End-to-end orchestration.** The framework attempts to cover the full stack: high-level textual planning, storyboard/keyframe generation, per-shot video synthesis, subtitle generation, and audio output.
3. **Some self-awareness about evaluation limits.** The authors acknowledge that standard image/video metrics like FID/FVD are not directly applicable without ground truth, and they note differences between automated scoring and human preference.

**Weaknesses:**

1. **Continuity and coherence across shots are not convincingly addressed.**

    The method decomposes a movie into many short, independently generated shots. However, the paper does not provide a clear mechanism to guarantee *visual continuity* across these shots — e.g., consistent character appearance, environment style, motion direction, lighting, or camera grammar. Section 3.2.3 claims that the *Shot Plan Agent* is designed to enforce visual continuity, but the description is vague, and Equation (4) plus the proposed heuristics do not actually demonstrate how separate clips are merged into a coherent multi-shot sequence without stylistic breaks.

    Similarly, while the system claims to produce accompanying audio and subtitles, the audio is generated separately from the video content. The paper does not explain how lip motion, speech timing, or emotional delivery are synchronized with the generated footage. Without actual AV alignment, the system is effectively still a shot-level video generator plus an unrelated audio track, rather than an integrated long-form audiovisual generator.

    These two issues cut to the core claim of “automated long-form movie generation.” As presented, the method behaves more like a collection of disconnected short clips. The evaluation reflects this: despite the stated goal of *multi-scene long-form narrative generation with character and plot consistency*, most reported metrics are **per-shot VBench** scores. These metrics capture short-clip qualities such as subject/background consistency, motion smoothness, and aesthetics, but not *movie-level* structure (story causality, character arcs, act/beat transitions, etc.). The ablations (CoT on/off, number of agents, LLM choice) are also analyzed only at the shot level on a very small test set. There is no quantitative measure of cross-scene identity persistence, narrative causality, or structural coherence across acts. In practice, the proposed method only slightly improves “narrative coherence,” and even that claim rests on a subjective two-rater study rather than robust movie-scale metrics.

2. **Evaluation design is narrow and potentially biased.**

    The benchmark (**MoviePrompts**) consists of just 10 prompts, 8 of which are adapted from very well-known IP (e.g., *Ne Zha 2, Frozen II, Inside Out 2*). These prompts are authored or curated by two annotators, and the human evaluation is performed by only two expert raters. This is (i) too small for statistical reliability, (ii) not blinded, and (iii) highly vulnerable to data contamination, because modern LLMs and video generators have been widely exposed to these franchises during training. In other words, the system may be doing style imitation or recall rather than true generalization.

3. **Metrics do not match the paper’s central claim.**

    The work claims advances in long-range narrative coherence, character consistency across scenes, and audio–subtitle synchronization. But the reported gains are mostly in per-shot VBench categories (e.g., Subject/Bg Consistency, Motion Smoothness, Dynamic Degree, Aesthetic Quality). These do not measure:

    - whether the same character looks and behaves the same in Scene 1 vs. Scene 7,
    - whether dialogue and events obey causal structure over time,
    - whether subtitle timing matches speech,
    - or whether conversations across scenes remain semantically consistent.

    In short, the metrics validate “this looks like a decent short clip,” not “this is a coherent movie.”

4. **Unsubstantiated cost/efficiency claims.**

    The paper frames the system as providing “near-zero cost” automated filmmaking.  Long-form video generation with multiple diffusion/transformer calls per shot, plus iterative LLM planning, is computationally and financially expensive. The paper does not report GPU hours, token usage, inference latency, or dollar cost per minute of final output. Without that, the cost claim may not stand.

5. **Audio evaluation is underspecified.**

    The method claims synchronized subtitles and stable audio generation. However, there is no quantitative evidence of audiovisual synchronization.  As a result, the audio claims are qualitative and not verifiable.

**Questions:**

1. **Data contamination & fairness.**

    Since MoviePrompts heavily relies on famous franchises (*Frozen II*, *Inside Out*, *Ne Zha*, etc.), how do you control for prior exposure of the LLMs and video generators to these IPs? Can you show results on held-out, original scripts with entirely novel characters to rule out simple style imitation or memorization?

2. **Movie-level narrative coherence.**

    Do you compute any structured, sequence-level metrics (e.g., story graph consistency, beat/act structure adherence, causal entailment across scenes)? If not, how do you justify claims of “coherent multi-scene narrative” beyond two human raters’ impressions?

3. **Audio / subtitle synchronization.**

    How is audiovisual sync actually enforced? Is there any quantitative AV-sync metric demonstrating that the generated speech matches the generated visuals?

4. **Cross-scene character identity and continuity.**

    What concrete mechanism ensures that the same character maintains consistent visual identity, style, and personality traits across different scenes and shots?

5. **Cost and latency.**

    What is the actual cost per minute of finalized output, including failed trials and regeneration passes? Please report GPU hours, dollar estimate, and LLM token usage per finished minute.

6. **Ablation granularity.**

    The paper reports a small numerical gain when enabling Chain-of-Thought reasoning (e.g., avg. score improves from 3.55 to 3.61). Which part of the CoT is responsible? Role decomposition (Director vs. Scene vs. Shot agent)? A more fine-grained ablation would make the contribution clearer.

---

> ### Author Response · Authors · 2025-11-18
> **Response (Part 1)**
>
> We thank the reviewer for the constructive feedback and valuable suggestions.
>
> **Q1**:*Data contamination & fairness.*
>
> **A1**: We appreciate the reviewer’s concern regarding potential prior exposure of LLMs or video generators to well-known IPs. We want to highlight two perspectives:
> - **LLM (script reasoning)**. Among the 10 movie scripts used in our experiments, two are entirely original, self-created scripts, specifically designed to test generalization and reduce memorization risk. Furthermore, certain movie scripts, such as Ne Zha 2, were released in January 2025, while our experiments used GPT-4o (2024-08-06), ensuring that the model could **not have seen** the script during training.
> - **Visual generator.** Our framework is based on Stable Diffusion (2022), whose training data predates these IPs (Ne Zha 2), meaning that the visual models could not have accessed related movie content or scripts.
>
> To further validate fairness, we also introduced two newly designed movie script settings to test performance on entirely novel plots and characters:
>
> | Characters |  Script Synopsis  |  Visual Appeal  | Script Faithfulness | Character Consistency | Physical Law | Narrative Coherence |
> |-------------|----------------|------------------------|------------------------|------------------------|------------------------|------------------------|
> | Iron Man;  Nezha; Spider-Man; Wukong | When a mysterious force threatens to collapse multiple realities, Iron Man, Nezha, Spider-Man, and Wukong unite to stop the disaster. Iron Man uses his technology to trace the source, while Nezha and Wukong wield their divine powers to contain it, and Spider-Man’s agility and quick thinking help keep chaos under control.      | 3.83 | 3.67  | 3.91    | 3.12    | 3.40 |
> | Tom Cruise;  Emma Watson | When a failed time experiment threatens to erase history, physicist Dr. Evelyn Cross (Emma Watson) and time agent Cole Turner (Tom Cruise) must travel across fractured timelines, from ancient Rome to a cyberpunk future, to restore reality before it’s lost forever.   | 3.92 | 3.78  | 3.97    | 3.41    | 3.39 |
>
>
> **Q2**:*Movie-level narrative coherence.*
>
> **A2**: We appreciate the reviewer’s question. We additionally employ structured, sequence-level metrics from ViStoryBench[2] to quantitatively evaluate multi-scene narrative coherence.
>
> As shown below, MovieAgent achieves state-of-the-art performance across all key metrics, including Scene Score, Cross-Character Consistency, Global Character Action Score, and Single Character Action Score:
>
> | Aspect                | Scene Score | Cross Consistancy of Character  | Global Character Action Score | Single Character Action Score  |
> |------------------------|----------------|---------------------|------------------------|------------------------|
> | StoryGen        | 0.90             | 37.5                 | 1.24                   | 1.48           |
> | StoryDiffusion | 1.88            | 37.7                | 2.41                   | 2.23           |
> | Vlogger[1] | 1.60            |  33.4                | 2.20                  | 2.07           |
> | MovieAgent | **3.46**           |   **38.4**                | **3.23**                  | **2.50**           |
>
> These metrics explicitly measure scene-level structural coherence and cross-scene causal consistency, providing more rigorous evidence than subjective impressions.
>
> [1] Vlogger: Make your dream a vlog. In CVPR, 2024
>
> [2] Zhuang C, Huang A, Cheng W, et al. Vistorybench: Comprehensive benchmark suite for story visualization[J]. arXiv preprint arXiv:2505.24862, 2025.

---

> > ### Author Response · Authors · 2025-11-18
> > **Response (Part 2)**
> >
> > **Q3**:*Audio / subtitle synchronization.*
> >
> > **A3**: Thank you for the thoughtful question. We clarify how MovieAgent enforces audio–video synchronization and how it is quantitatively evaluated.
> > - **How synchronization is enforced.** In MovieAgent, since the system follows a script → scene → shot → dialogue hierarchy, the semantic alignment between speech, subtitles, and visual actions is structurally enforced by design.
> > - **Why standard AV-sync metrics cannot be applied.** Script-to-movie generation differs fundamentally from traditional talking-head settings[1][2]. The number and duration of shots are not **predetermined**. There is **no ground-truth** video–audio alignment for the generated movie. Therefore, conventional metrics such as PSNR, FID, or lip-sync error used in speech-driven animation cannot be directly used.
> > - **Alternative quantitative evaluation**. To address this, we provide a cross-modal alignment metric that measures semantic and temporal consistency between video and audio embeddings. Specifically, Sync-C[3] and Sync-D[4] are used for evaluation. The corresponding results are summarized in the table below:
> >
> > | Metric | Score  | Description |
> > |--------|---------|-------------|
> > | Sync-C[3] | 7.052   | Measures semantic alignment between lip motion and audio (higher is better) |
> > | Sync-D[4] | 7.667   | Measures temporal smoothness of lip dynamics (lower is better) |
> >
> > In fact, MovieAgent achieves competitive performance on both Sync-C and Sync-D, as shown in the comparison in Table 2 of Hallo2 [5]. Although the test sets differ, the results still serve as a meaningful reference.
> >
> > [1] Tan, Shuai, Bin Ji, Mengxiao Bi, and Ye Pan. "Edtalk: Efficient disentanglement for emotional talking head synthesis." In European Conference on Computer Vision, pp. 398-416. Cham: Springer Nature Switzerland, 2024.
> >
> > [2] Ma, Yifeng, Suzhen Wang, Yu Ding, Bowen Ma, Tangjie Lv, Changjie Fan, Zhipeng Hu, Zhidong Deng, and Xin Yu. "Talkclip: Talking head generation with text-guided expressive speaking styles." IEEE Transactions on Multimedia (2025).
> >
> > [3] Prajwal, K. R., Rudrabha Mukhopadhyay, Vinay P. Namboodiri, and C. V. Jawahar. "A lip sync expert is all you need for speech to lip generation in the wild." In Proceedings of the 28th ACM international conference on multimedia, pp. 484-492. 2020.
> >
> > [4] Wei, Cong, Bo Sun, Haoyu Ma, Ji Hou, Felix Juefei-Xu, Zecheng He, Xiaoliang Dai et al. "Mocha: Towards movie-grade talking character synthesis." arXiv preprint arXiv:2503.23307 (2025).
> >
> > [5] Cui, Jiahao, Hui Li, Yao Yao, Hao Zhu, Hanlin Shang, Kaihui Cheng, Hang Zhou, Siyu Zhu, and Jingdong Wang. "Hallo2: Long-duration and high-resolution audio-driven portrait image animation." arXiv preprint arXiv:2410.07718 (2024).
> >
> > **Q4**:*Cross-scene character identity and continuity.*
> >
> > **A4**: Thank you for the question. MovieAgent ensures cross-scene character consistency through a concrete mechanism based on Mix-of-Show’s ED-LoRA[6]. For each character, we train a dedicated LoRA embedding that learns the character’s visual identity, style, and personality traits. During generation, this character-specific LoRA is consistently applied in every scene and shot, ensuring that the same identity is preserved across different locations, lighting conditions, and narrative contexts.
> >
> > [6] Gu, Yuchao, Xintao Wang, Jay Zhangjie Wu, Yujun Shi, Yunpeng Chen, Zihan Fan, Wuyou Xiao et al. "Mix-of-show: Decentralized low-rank adaptation for multi-concept customization of diffusion models." Advances in Neural Information Processing Systems 36 (2023): 15890-15902.
> >
> > **Q5**:*Cost and latency.*
> >
> > **A5**: Thank you for the question. We provide detailed cost estimates for training, inference cost:
> > - **GPU hours and dollar cost**: Both the GPU cost and the LLM API cost of MovieAgent are very low, making the system practical and affordable. Details are shown in the table below:
> >
> > Table 1. GPU Hours and Dollar Cost on Frozen II
> > | Component                | Training Consumption (s)  | Inference Consumption (s)  | Dollar Cost  | Number |
> > |------------------------|----------------|------------------|-----------------------|-----------------------|
> > | Director Agent (gpt-4o)    | - | 13.48 s       | ~$0.0102 | 6 sub-scripts|
> > | Scene Plan Agent (gpt-4o)    | - | 68.98 s      | ~$0.0304 | 6 sub-scripts, 19 scenes|
> > | Shot Plan Agent (gpt-4o)        | -   |169.10 s    | ~$0.354 | 6 sub-scripts, 19 scenes, 59 shots|
> > | Image Generation  (ROICtrl)  | 7080.00 s (5 characters) | 562.98  s    | -           | 6 sub-scripts, 19 scenes, 59 shots|
> > | Video Generation (SVD) | - | 3413.25  s     | -       | 6 sub-scripts, 19 scenes, 59 shots|
> > | Total (~60 shots) | 7080.00 s (≈ 118.00 mins) | 4227.79 s (≈ 70.46 mins)   | ~$0.3946      | On 1×A6000 GPUs|
> > | Total (~60 shots) | 1416.00 s (≈ 23.6 mins) | 685.23 s (≈ 11.42 mins)     | ~$0.3946   | On 8×A6000 GPUs, (parallel)|

---

> ### Author Response · Authors · 2025-11-18
> **Response (Part 3)**
>
> **Q6**:*Ablation granularity.*
>
> **A6**: Thank you for the suggestion. First, we clarify that the average score does not reflect the impact of CoT, since metrics like Visual Appeal and Physical Law are mainly determined by the visual generator. Narrative Coherence is the only dimension that meaningfully captures the effect of CoT. We conducted fine-grained ablations using the provided five variants (no-CoT, Director-CoT, Scene-CoT, Shot-CoT, full-CoT) for narrative coherence metric. The results show clear functional distinctions:
>
> Table 1. The Ablation for CoT
> | CoT                | Narrative Coherence | Example (Details) |
> |------------------------|----------------|----------------|
> | -   | 3.09  | |
> | Director-CoT (gpt-4o)    | 3.10  | **Without CoT**: Sub-Script 1 begins abruptly: “Anna, Elsa, Kristoff, Olaf, and Mattias embark on a journey…”.  **Director-CoT**:rewrites the opening into a coherent setup: “Anna and Elsa discuss the mysterious voice calling Elsa…”. **Conclusion**: Director-CoT enhances the high-level plot outline, ensuring motivations (*mysterious voice*), world events, and relationships are consistently established across sub-scripts.|
> | Scene-CoT (gpt-4o)    | 3.13 | **Without CoT**: scene order frequently jumps: Forest to Castle with no linking transitions. **With Scene-CoT**: Producer-level transitions appear naturally: “They arrive at the forest…”. **Conclusion**: Scene-CoT organizes events into logically ordered scenes with smoother transitions. |
> | Shot-CoT (gpt-4o)        | 3.22   | **Without-CoT**: Wrong-character appearances (“Kristoff speaks to Elsa” while only Olaf is in the shot). **With Shot-CoT**: Shots correctly reflect the characters present. **Conclusion**: Shot-CoT improves visual fidelity by aligning shot-level details with scene goals and characters.|
> | Director-CoT & Scene-CoT & Shot-CoT  | 3.29 |  Full-CoT is the only setting where three levels (script, scene, shot) remain fully aligned, producing coherent multi-level reasoning|
>
> The results show that each CoT component contributes incrementally, with Shot-CoT providing the largest single gain and Full-CoT achieving the highest Narrative Coherence (3.29) through fully aligned script, scene, and shot reasoning; the detailed evaluation protocol can be found in the supplementary material.

---

### Meta-Review · Area_Chair_CbJn · 2026-01-06

**Summary:**

The AC looked at the movies in the supp material qualitatively and is not convinced by the capability of MovieAgent. Moreover, the AC has the same concern with the reviewers that the benchmark is very small, 10 movies. The authors did add two more in the rebuttal and claim that the total video seconds are not small, but the AC thinks that we are talking about movie so the unit is per movie.

**Reviewer Concerns:**

Dataset size and quality of the movies generated are not addressed.

**Reviewer Scores:**

AC doubt that the reviewers' scores will improve after rebuttal.

---

### Decision · Program_Chairs · 2026-01-26

Reject